# Development and Validation of a HPLC–MS/MS Method to Measure Nifuroxazide and Its Application in Healthy and Glioblastoma-Bearing Mice

**DOI:** 10.3390/pharmaceutics14102071

**Published:** 2022-09-28

**Authors:** Tommaso Ceruti, Quintino Giorgio D’Alessandris, Roberta Frapolli, Jay Gopalakrishnan, Mariachiara Buccarelli, Marina Meroni, Liverana Lauretti, Lucia Ricci-Vitiani, Roberto Pallini, Massimo Zucchetti

**Affiliations:** 1Laboratory of Cancer Pharmacology, Department of Oncology, Istituto di Ricerche Farmacologiche Mario Negri IRCCS, 20156 Milan, Italy; 2Department of Neurosurgery, Fondazione Policlinico Universitario A. Gemelli IRCCS, 00168 Rome, Italy; 3Institute of Human Genetics, University Hospital Düsseldorf, Heinrich-Heine-Universität, Moorenstr. 5, 40225 Düsseldorf, Germany; 4Department of Oncology and Molecular Medicine, Istituto Superiore di Sanità, 00161 Rome, Italy; 5Institute of Neurosurgery, School of Medicine, Catholic University Rome, 00168 Rome, Italy

**Keywords:** nifuroxazide, liquid chromatography–tandem mass spectrometry technique, pharmacokinetics, murine model, drug repositioning

## Abstract

Nifuroxazide (NAZ), a nitrofuran derivative used to treat diarrhea, has been recently shown to possess anticancer activity. However, its pharmacokinetic profile is poorly known. The pharmacokinetic profile of NAZ was thus investigated in mice using a newly developed method based on high-performance liquid chromatography–tandem mass spectrometry (HPLC–MS/MS). We determined the concentrations of NAZ in the plasma and brain tissue of mice treated with the drug. The method proved to be specific, reproducible, precise, and accurate. It also demonstrated high sensitivity, reaching an LOQ in the order of ppb for both matrices, using samples of 100 µL or 0.2 g. The new HPLC–MS/MS assay was successfully applied to study the pharmacokinetics of NAZ after chronic intraperitoneal administration in mice at a dose of 30 mg/kg. One hour after treatment, plasma concentrations of NAZ were in the range of 336–2640 ng/mL. Moreover, unlike the brains of healthy mice or those with healed mechanical injuries, we found that NAZ was able to cross the injured blood–brain barrier of tumor-infiltrated brains. Thus, following i.p. administration, NAZ reaches systemic levels suitable for testing its efficacy in preclinical models of glioblastoma. Overall, these pharmacokinetic data provide robust evidence supporting the repositioning of NAZ as an antitumor drug.

## 1. Introduction

Nifuroxazide (NAZ) is a nitrofuran derivative (4-hydrossi-N′-[(5-nitrofuran-2-il) metilene] benzoidrazide) with a broad spectrum of antibacterial activity, and has been in use since the 1960s for the treatment of infectious diarrhea [1,2]. In 2008, the ability of NAZ to potently inhibit the protein-coding gene *STAT3* (signal transducer and activator of transcription 3) was discovered, suggesting its potential use as an anticancer agent [3]. This new pharmacological property was investigated in 2015, when several studies highlighted the anticancer potential of NAZ and its ability to cause apoptosis in cancer cells [4,5], supporting the repositioning of the drug as a targeted anticancer agent.

NAZ is used mainly as an oral drug, given its prevalent local use as an antiseptic agent on the mucous membrane of the intestinal tract [1,2]. In the scientific literature, information on the pharmacokinetic profile of NAZ is scarce; in particular, there have been no studies investigating its distribution in tissues and tumors in detail. A sensitive method that is suitable for multiple biological matrices is required to exert such investigation. Since only an outdated method was available in the literature—one that is not highly sensitive and requires a large quantitative matrix [6]—we created and validated a new quantitative assay using high-performance liquid chromatography coupled with tandem mass spectrometry (HPLC–MS/MS) to measure NAZ in the plasma and brain tissue of mice. The study was conducted on healthy mice, mice grafted with patient-derived glioblastoma stem-like cells (GSCs), and sham-operated controls. Our method measured NAZ with high precision and accuracy, demonstrating for the first time the capacity of the drug to distribute in brains with tumors.

## 2. Materials and Methods

### 2.1. Reagents and Chemicals

The analytical reference standard of NAZ (batch no: BCBW4413) was purchased from Sigma-Aldrich (Saint Louis, MO, USA), while the reference standard of its deuterated derivative [^2^H_4_]-nifuroxazide (batch no: J0720)—used as an internal standard (IS)—was obtained from Santa Cruz Biotech (Dallas, TX, USA). HPLC-grade acetonitrile (CH_3_CN) and ammonium acetate (CH_3_COONH_4_) and analytical-grade 99% formic acid (HCOOH) were purchased from Carlo Erba (Milan, Italy). The Milli-Q purification system, used to produce deionized water, was purchased from Millipore Corp. (Bedford, MA, USA).

### 2.2. Tumor Cell Line and Animals

Glioblastoma stem-like cell line #1 (GSC#1) was isolated from surgical samples of an adult patient who underwent craniotomy at the Institute of Neurosurgery, Catholic University of Rome, upon approval by the local ethical committee (Prot. ID 2253). Informed consent was obtained from the patient before surgery. After mechanical dissociation, single-cell suspensions were cultured in a serum-free medium supplemented with epidermal growth factor (EGF) and basic fibroblast growth factor (b-FGF), as previously described [7,8]. The GSC#1 line was validated by short tandem repeat (STR) DNA fingerprinting. Nine highly polymorphic STR loci plus amelogenin (Cell ID^TM^ System, Promega Inc., Madison, WI, USA) were used [9]. The GSC#1 profile was challenged against public databases to confirm its authenticity [9]. The in vivo tumorigenic potential of GSC#1 was assayed by intracranial cell injection into immunocompromised mice, resulting in tumors with the same antigen expression and histological tissue organization as the parent tumor [8,10].

### 2.3. Preparation of Standard and Quality Control Solutions

Two different aliquots of NAZ powder were weighed separately and then dissolved in DMSO to obtain two independent 1 mg/mL mother solutions for the preparation of standards and quality controls. A set of working solutions (WSs), in the range 5.0–1000.0 ng/mL, was then prepared by diluting the mother solutions with absolute ethanol. The WSs were used to prepare the standard points of the calibration curve and the quality control (QC) samples in plasma and brain tissue. Moreover, the stock solution for the IS was prepared in DMSO at 1.0 mg/mL and, subsequently, diluted in ethanol to obtain WSs at 100.0 ng/mL for plasma and brain samples. All types of standard solutions were stored at −20 °C.

### 2.4. Preparation of Standard and QC Brain Samples

The brains of control mice were weighed and added to CH_3_COONH_4_ at pH 4.5, at a 1:3 ratio (*w*/*v*). They were homogenized by Ultra Turrax in polypropylene tubes, and then 20 µL of each WS was mixed with 180 µL of homogenate in a polypropylene tube to obtain the standards for the calibration curve and the QC samples. The final NAZ standard concentrations achieved in the brain homogenates were as follows: 0.5, 1.0, 2.5, 5.0, 10.0, and 20.0 ng/mL, corresponding to a range of 2.0–80.0 ng/g. The QCs were obtained at 1.5, 7.5, and 15.0 ng/mL, corresponding to 6.0, 30.0, and 60.0 ng/g, respectively.

### 2.5. Preparation of Standard and QC Plasma Samples

To prepare the standards for the calibration curves and the QCs, 90 µL aliquots of plasma were combined with 10 µL of the matching WS. The final NAZ concentrations for the standard samples were 1.0, 2.5, 5.0, 10.0, 20.0, and 100.0 ng/mL, while they were 3.0 and 75.0 ng/mL for the QCs.

### 2.6. Processing Plasma and Brain Samples

In the preliminary phase, we tried different solvents for extraction—2-propanol, methanol, and acetonitrile—at different pH conditions, quickly obtaining a satisfactory recovery with acetonitrile (CH_3_CN) mixed with 0.1% HCOOH. Briefly, the biological matrices were processed as follows: Murine plasma samples were thawed at room temperature, and 100 µL study samples, standards, and QCs were transferred to polypropylene Eppendorf tubes and mixed with 10 µL of IS working solution. The samples were then mixed with 400 µL of CH_3_CN containing 0.1% HCOOH and vortexed to deproteinize the plasma. The mixture was then centrifuged at 15,900× *g* and 4 °C for 10 min. The upper organic phase was transferred to a new polypropylene Eppendorf tube, dried under a gentle nitrogen flow at 40 °C, and dissolved with 100 µL of a mixture (1:1 *v*/*v*) of mobile phases A and B. The samples were then centrifuged for 10 min at 15,900× *g* and 4 °C. The obtained supernatant was transferred into glass vials, and 5 µL was injected and analyzed by HPLC–MS/MS. The brain samples were weighed and mixed with three volumes (*w*:*v*) of CH_3_COONH_4_ at pH 4.5, before being processed in the same way as the control homogenate. Then, 200 µL study samples, standards, and QC samples were transferred to polypropylene tubes, combined with 20 µL of IS working solutions to obtain a final concentration of 10 ng/mL, and mixed. The samples were deproteinized with 1 mL of CH_3_CN containing 0.1% HCOOH, vortexed, and then centrifuged at 15,900× *g* for 10 min at 4 °C. The organic upper phase was transferred to an Eppendorf tube, dried under nitrogen, reconstituted with 80 µL of the mobile phase mixture, and then processed and analyzed as described for plasma samples.

### 2.7. Liquid Chromatography

A series 200 HPLC system (PerkinElmer, Waltham, MA, USA) equipped with two micropumps and an autosampler—both linked to a vacuum degasser and a temperature-controlled column compartment—was used to perform the chromatographic analysis. Samples were separated at 30 °C on a 50 mm × 2.0 mm Gemini C_18_ column with 5 µm particle size, coupled with a Security Guard™ ULTRA cartridge made of the same material provided by Phenomenex (Torrance, CA, USA). The mobile phase (MP) was prepared by adding 0.1% HCOOH in double-distilled water (MP A) and in acetonitrile (MP B). The system flow rate was set up at 0.2 mL/min, and the MP was fluxed according to the following gradient steps: (1) 80% MP A held for 2 min; (2) from 80% to 5% MP A in 3 min; (3) 5% MP A held for 1 min; and (4) return to the starting condition in 1 min and re-equilibration for 5 min, for a total run time of 12 min. The autosampler was maintained at 4 °C throughout the analysis.

### 2.8. Mass Spectrometry

An API 4000 triple-quadrupole mass spectrometer (SCIEX, Framingham, MA, USA) outfitted with a turbo ion spray source set at 450 °C and 4500 V was used for the mass spectrometric identification of NAZ. Electrospray ionization (ESI) was performed in positive ion mode with a needle current of 4 µA and used for the analysis of biological samples. The nebulizer gas (Gas 1) and heater gas (Gas 2) consisted of purified air, while nitrogen was used as the curtain gas and the collision-activated dissociation gas. These gases were maintained at 40, 50, 30, and 5 instrument units (psi), respectively. The declustering potential (DP) was set at 96 V and the collision exit potential (CXP) at 11 V. Direct infusion of 1 µg/mL standard solutions of NAZ and IS in acetonitrile was used to set all of the parameters of the source. The final quantification was carried out in selected reaction monitoring (SRM), with the following transitions: m/z 276.0→121.2 (collision energy of 27 eV) for NAZ and m/z 280.2→115.0 (collision energy 52 eV) for the IS. Figure 1 shows the patterns of fragmentation, MS, and MS/MS mass spectra of all molecules analyzed. The software package Analyst 1.6.2 (AB SCIEX) was used for the data processing.

### 2.9. Validation of the Method

The validation of the bioanalytical procedures was carried out in compliance with the US Food and Drug Administration and European Medicines Agency criteria [11,12]. The NAZ assay that we developed for plasma and brain tissue is a new method validated in terms of recovery, lower limit of quantitation (LLOQ), matrix effect and selectivity, linearity of the calibration curve, intraday and interday precision, and accuracy and stability.

Only one previously existing method for assessing NAZ levels in plasma was found in the relevant literature [6]. Using liquid chromatography coupled with mass spectrometry, we obtained a highly sensitive and reliable method that was further enhanced by the extraction step, which used small volumes of sample (100 µL instead of 2 mL), acetonitrile as a deproteinizing agent instead of chloroform for solvent extraction, and the addition of deuterated NAZ instead of nifuratel as the IS.

### 2.10. Tumor Model and Pharmacokinetic Study

Experiments involving animals were authorized by the Italian Ministry of Health (n° 144/2021-PR). The pharmacokinetic study was first conducted in healthy, 4−6-week-old male SCID mice (Charles River, Calco, Italy). The mice were treated intraperitoneally (i.p.) with NAZ dissolved in DMSO at a dose of 15 mg/kg, 3 times/week, for two weeks. Blood was collected at pre-dose and at 1, 3, and 6 h after the last treatment. Three mice were sampled at each time point. Another three mice were injected with the vehicle and sampled 1 h later to obtain plasma and brain samples as blank control tissues. The animals were anesthetized with isoflurane, and their blood was drained from the retro-orbital plexus into heparinized tubes and then centrifuged for 10 min at 4000 rpm and 4 °C to separate the plasma. After euthanasia, the brain was quickly removed, frozen on dry ice, and stored at −80 °C together with the plasma samples until analysis by HPLC–MS/MS. For a parallel study, we collected blood and brain tissue from a limited number of mice bearing intracranial GSC#1 tumors, or sham-operated to evaluate the impact on drug distribution caused by BBB alterations due to tumor growth and/or surgery. For brain grafting, mice were intracranially implanted with GSC#1 tumor cells (2 × 10^5^ GSC # 1) expressing the GFP protein, suspended in 5 μL of serum-free medium. Mice were anesthetized via i.p. injection of diazepam (2 mg/100 g) followed by intramuscular injection of ketamine (4 mg/100 g). The animals’ skulls were immobilized in a stereotactic head frame, and a burr hole was made 2 mm right of the midline and 1 mm anterior to the coronal suture, after which cells were slowly injected using the tip of a 10 μL Hamilton microsyringe placed at a depth of 3 mm from the dura. Sham-operated control mice (*n*, 3) underwent needle penetration and saline injection (5 μL) into the brain using the same technique. Taking into account the fact that the blood–brain barrier (BBB) requires around 3–4 weeks to repair after mechanical damage caused by implantation [13], the mice were treated starting 1 week after surgery, for 3 weeks, by systemic injection of NAZ dissolved in DMSO. Four weeks after surgery and 1 h after the last administration of NAZ, the mice were deeply anesthetized, and whole blood samples were withdrawn into heparinized tubes and promptly processed to obtain plasma. After euthanasia, the brain was quickly removed and stored at −80 °C in liquid nitrogen together with the plasma samples until analysis by HPLC–MS/MS.

## 3. Results

### 3.1. HPLC–MS/MS

Sample preparation was simple and straightforward. It involved quick protein precipitation and gave satisfactory results of HPLC–MS/MS analysis, showing good sensitivity in plasma and brain homogenates with a small volume of sample. The extracted plasma and brain samples’ SRM chromatographic patterns are shown in Figure 2 and Figure 3, respectively. NAZ and IS were both eluted after 3.1 min, with no presence of interfering peaks visible at this retention time. Double-blank, blank, and LLOQ samples at 1.0 ng/mL for plasma and at 2.0 ng/g for brain tissue are displayed in panels A, B, and C, respectively. Panel D of Figure 2 and Figure 3 shows samples of plasma and GSC#1-tumor brain tissue, respectively, collected from a mouse 1 h after the last NAZ treatment. The mean concentrations of NAZ measured were 2330 ng/mL in plasma and 5.8 ng/g in brain tissue with tumor.

### 3.2. Recovery

The recoveries were 81.9% (CV 21.3%) for QCL and 92.8% (CV 5.1%) for QCH in plasma, and 100.8% (CV 7.2%) for QCL and 98.8% (CV 7.9%) for QCH in brain tissue, based on five replicates at low and high QC concentrations for each matrix. In plasma and brain tissue, the recovery for the IS was 85.8% and 104.1%, respectively.

### 3.3. LLOQ Selectivity and Matrix Effect

We initially tried to validate the LLOQ at 0.5 ng/mL and 1.0 ng/g for plasma and brain tissue, respectively, but failed to achieve a satisfactory signal-to-noise-ratio (S/N > 10) as requested by the FDA and EMA guidelines.

More than satisfactory results were obtained by validating the LLOQ at the concentrations of 1.0 ng/mL (S/N = 19.2) for plasma and 2.0 ng/g (S/N = 17.5) for brain tissue. The precision and accuracy results for plasma were 13.2% and 97.6%, respectively, while they were 4.8% and 108.3% for brain tissue, respectively. No matrix effect was observed from spiking six independent sources of blank murine plasma with NAZ at concentrations corresponding to the lower and higher QCs, with a precision of 2.0% and 4.4%, respectively. Similar results were obtained in brain homogenates, with precision of 1.9% and 2.7% at the lower and higher QCs, respectively. The method proved to be selective for the analyte; in fact, no co-eluting peak of substances of the matrices affected the NAZ signal, with the blank noise being negligible (<6%) at the LLOQ for both plasma and brain tissue.

### 3.4. Linearity

NAZ calibration curves in plasma and brain tissue were prepared on different days during the validation process. The results are presented in Table 1 and Table 2 for plasma and brain tissue, respectively. The standard curves tested proved to be linear in the evaluated concentration range for both matrices. The back-calculated standards’ concentration accuracy was ≥97.1 and ≥91.8%, with precision expressed as CV ≤ 4.6% and ≤6.4%. The Pearson’s coefficient of determination, R^2^, was ≥0.995 for both the plasma and brain tissue calibration curves. The carryover effect was evaluated on a blank sample injected after the upper limit of quantitation. No carryover effect was observed for NAZ or IS upon injecting an extracted blank sample after the upper limit of quantitation.

### 3.5. Accuracy and Precision

Five distinct replicates of QC samples at 3.0, 15.0, and 75.0 ng/mL for plasma and 6.0, 30.0, and 60.0 ng/g for brain tissue were analyzed in a single-run analysis to evaluate intraday accuracy and precision. The method was highly accurate and precise; in fact, the accuracy ranged from 90.2 to 94.8% and the precision from 4.0 to 6.4% in plasma; the same metrics were in the ranges 105.4–111.8% and 1.7–3.3%, respectively, in brain tissue, as shown in Table 3, Table 4, Table 5 and Table 6. Similar results were obtained when the interday variability was assessed. The precision and accuracy determined in triplicate samples of each QC level, over the course of three days of analysis (five for plasma), displayed ranges of 9.8–11.6% and 97.2–98.6% in plasma, respectively, and 3.5–4.7% and 106.4–110.5% in brain tissue, respectively, as shown in Table 3 and Table 4.

### 3.6. Stability

We evaluated three replicates of QCs at 3.0 and 75.0 ng/mL for the stability of NAZ, after keeping them for 4 h at room temperature (RT) and, after extraction, for 48 h in the autosampler at 4 °C. The amounts recovered were 93.9% and 92.8%, and 111.7% and 108.9% of the nominal concentrations, respectively.

The stability of NAZ over time in the frozen samples was evaluated in plasma and in brain homogenates. After two months of storage at −80 °C, the drug was stable, as the concentrations remaining were equal to 93.8% and 98.6% of the nominal QC values for plasma, and close to 100% for brain tissue (QCs: 6 and 60 ng/g).

### 3.7. Application of the Method

Our method was suitable for determining the concentrations of NAZ in the plasma and brains of mice. Pharmacokinetic results were obtained in healthy control mice with intact brains, in mice with intracranially implanted GSC#1 tumor cells, and in sham-operated controls that underwent needle penetration with saline injection in the brain. As shown in Table 7, in healthy mice, 1 h after the last i.p. administration, NAZ achieved an average plasma concentration of 205 ng/mL, which was maintained after 3 h at 6.7 ng/mL. NAZ was detectable at 6 h at levels close the LLOQ, indicating rapid drug clearance with an apparent elimination half-life of 40 min. In the brain, at the same times, the drug was undetectable, suggesting that NAZ is unable to cross the intact BBB. In mice with orthotopic GSC tumors, 1 h after the last administration, NAZ achieved brain concentrations of 2.6 and 9.2 ng/g, while the drug was not detectable in any of the brains of the sham-operated mice. The NAZ plasma exposure of these mice ranged from 336 to 2640 ng/mL (Table 8).

Together, these findings demonstrate that the newly established method can accurately determine NAZ’s plasma disposition and its distribution in the brain, successfully supporting the study of its pharmacokinetics. They also show that chronic i.p. treatment allows the diffusion of NAZ into the brain across a tumor-damaged BBB.

## 4. Discussion

We developed and validated a new highly sensitive method to measure NAZ in biological matrices based on liquid chromatography–tandem mass spectrometry. Prior to this, only one method was available [6], based on HPLC analysis with limited sensitivity that entailed the extraction of NAZ by chloroform from plasma using large samples (2 mL) and injection (0.5 mL) volumes. We changed the extraction procedure and developed a new method to quantify NAZ via HPLC coupled with mass spectrometry. The procedure was validated for precision and accuracy, along with linearity of the calibration curve, along three run of analysis per matrix, in which we also established the recovery, specificity, and LLOQ. Under these conditions, we achieved a sensitivity of 1 and 2 ppb using samples of 100 µL or 0.2 g, respectively. The new method was applied successfully in a preliminary pharmacokinetic study of NAZ, measured in the plasma, brain tissue, and tumor-infiltrated brain tissue of mice receiving chronic i.p. treatment with the drug. We determined the mean levels of NAZ in the plasma and brain tissue of tumor-bearing mice to be 2330 ng/mL and 5.81 ng/g, respectively, concentrations corresponding to 8.5 µM and 0.02 µM, respectively. These results, although obtained with a limited number of mice, indicated that brain tumors—but not surgery per se—affect the permeability of the BBB, with NAZ being undetectable in the brains of healthy mice and sham-operated mice.

The clinically relevant levels of NAZ as antitumor drug are poorly known. No previously published data are available on the therapeutic dose of the drug—in vitro or in vivo—for brain tumors. In multiple-myeloma cell cultures, NAZ concentration of 3 µM could inhibit STAT3 activity [3]. Recently, it was shown that the half-maximal inhibitory concentration of NAZ in the HEL erythroleukemia cell line was 14 µM after 48 h of in vitro exposure [14]. Data from our group [15] showed that NAZ—similarly to multiple-myeloma cells—is effective in vitro on glioma stem cell lines at a concentration of 3 µM. In the present study, the peak plasma concentration of NAZ was 8.5 µM—a figure that supports the clinical usefulness of the drug. Moreover, we found that NAZ is able to cross the injured BBB and penetrate into the tumor, unlike in the brains of healthy mice or those with healed mechanical injuries, where the drug was undetectable. Moreover, there is robust evidence to support the repositioning of NAZ as an antitumor drug in brain cancer, as the recent literature suggests [16]. However, several issues remain to be solved in this path. First, the concentration that we found in brain tissue was 0.02 µM—a figure whose clinical relevance remains to be determined. Second, in the present study, NAZ was administered i.p. From a clinical viewpoint, parenteral administration could be desirable, but would require more extensive toxicity studies in vivo and a phase I trial. Modified-release formulations, or NAZ analogues with improved BBB penetration, could be developed *ad hoc.* To date, NAZ has been approved for clinical use only as an oral drug. Historical data show that after oral administration, NAZ undergoes poor absorption and, thus, achieves low plasma levels. However, these data have been criticized and, since nitrofurans are known to be prodrugs [17,18], it has been deduced that NAZ may undergo extensive metabolism. In fact, in rats, about 17% of orally administered NAZ at a dose of 10 mg/kg reached systemic circulation and was excreted in urine [19]. The effectiveness of oral NAZ in in vivo experiments supports this concept [17,20,21], suggesting that NAZ may be effective in a clinically relevant dose range. Crucially, in a murine model of melanoma [17], orally administered NAZ at 50–150 mg/kg—a dose similar to the one clinically administered for diarrhea—was able to eradicate melanoma cancer stem cells due to selective bioactivation by aldehyde dehydrogenase-1. Therefore, further studies are warranted to better assess the pharmacokinetics and anti-glioma role of orally administered NAZ.

In summary, we set up and validated a reliable method to assess NAZ levels in plasma and brain tissue, while also conducting a preliminary pharmacokinetic investigation. The application of the present method could foster preclinical studies aimed at drug repositioning of NAZ, and could also be successfully used for extensive pharmacokinetic investigations.

## Figures and Tables

**Figure 1 pharmaceutics-14-02071-f001:**
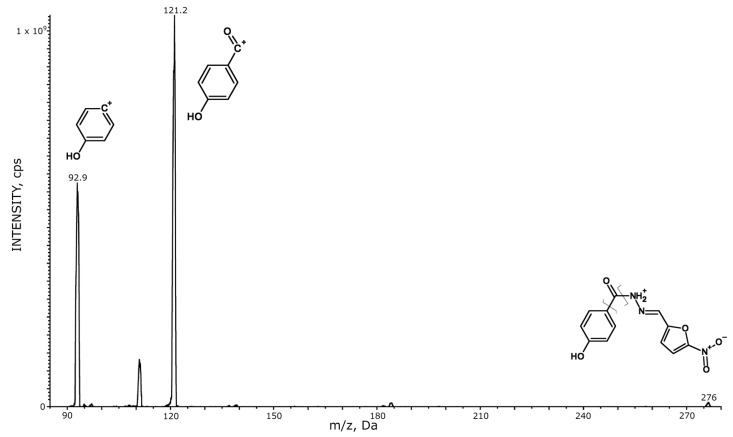
MS and MS/MS mass spectra for NAZ and IS ([^2^H_4_] nifuroxazide).

**Figure 2 pharmaceutics-14-02071-f002:**
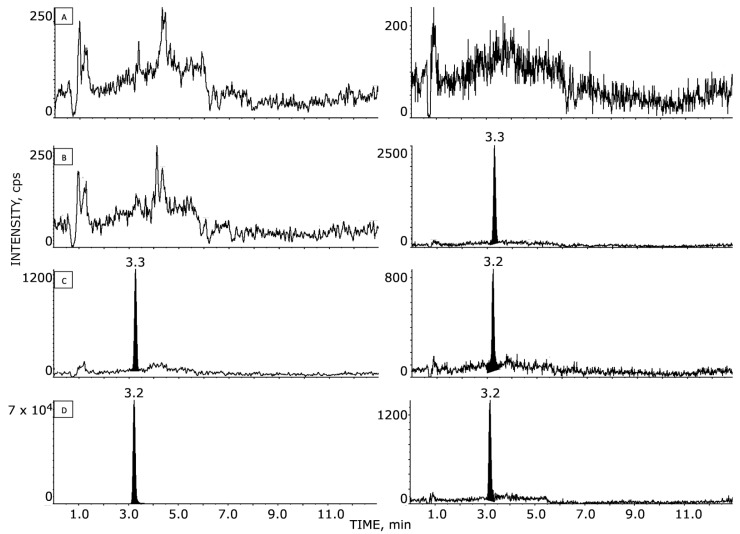
HPLC–MS/MS chromatograms of NAZ in murine plasma: (**A**) blank sample; (**B**) blank sample with added IS; (**C**) NAZ at the LLOQ (i.e., 1 ng/mL); (**D**) NAZ and IS of an extracted sample 1 h after the last drug administration; the measured concentration corresponds to 2330 ng/mL.

**Figure 3 pharmaceutics-14-02071-f003:**
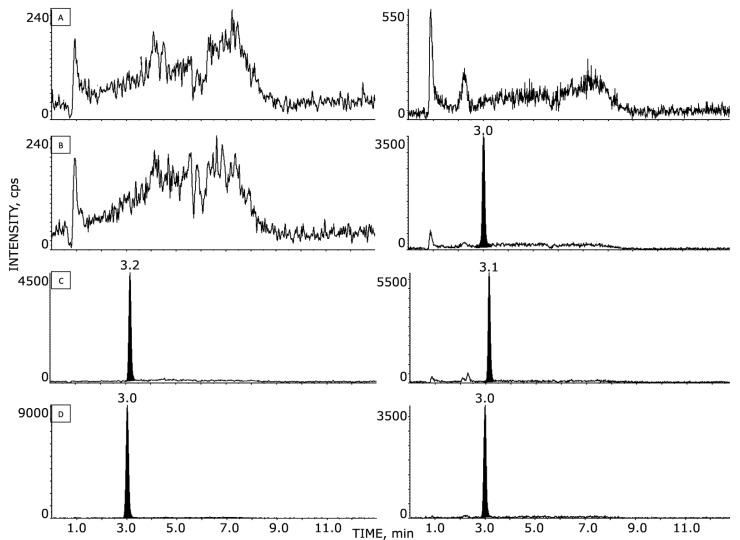
HPLC–MS/MS chromatograms of NAZ in murine brain tissue: (**A**) blank sample; (**B**) blank sample with added IS; (**C**) NAZ at the LLOQ (i.e., 2 ng/g); (**D**) NAZ and IS of an extracted sample 1 h after the last NAZ administration; the measured concentration corresponds to 9.2 ng/g.

**Table 1 pharmaceutics-14-02071-t001:** Linearity, accuracy, and precision of the NAZ calibration curve in plasma.

	Nominal Concentration (ng/mL)						
	1.0	2.5	5.0	10.0	20.0	100.0	
**NAZ**	Measured concentration						**R^2^ Calibration**
Day 1	1.1	-	5.1	10.1	19.6	94.5	0.996
Day 2	1.0	2.3	5.2	10.2	19.2	105	0.996
Day 3	1.0	2.4	5.2	10.1	19.2	103	0.998
Day 4	1.1	2.5	4.8	10.2	19.4	104	0.997
Day 5	1.0	2.5	4.7	10.2	21.1	97.2	0.996
**Mean (N = 5)**	**1.0**	**2.4**	**5.0**	**10.2**	**19.7**	**100.7**	**0.997**
**Accuracy (%)**	**103.8**	**97.1**	**99.6**	**101.6**	**98.5**	**100.7**	
**SD**	**0.04**	**0.1**	**0.2**	**0.05**	**0.8**	**4.6**	
**Precision (%)**	**3.8**	**4.6**	**4.5**	**0.5**	**4.1**	**4.6**	

**Table 2 pharmaceutics-14-02071-t002:** Linearity, accuracy, and precision of the NAZ calibration curve in brain tissue.

	Nominal Concentration (ng/g)						
	2.0	4.0	10.0	20.0	40.0	80.0	
**NAZ**	Measured concentration						**R^2^ Calibration**
Day 1	2.3	3.7	11.0	19.7	38.6	80.4	0.995
Day 2	2.0	3.8	10.0	20.6	40.5	78.9	0.993
Day 3	2.3	3.5	10.4	19.1	39.5	82.6	0.999
**Mean (N = 3)**	**2.2**	**3.7**	**10.5**	**19.8**	**39.5**	**80.6**	**0.996**
**Accuracy (%)**	**110.0**	**91.8**	**104.5**	**99.0**	**98.8**	**100.8**	
**SD**	**0.1**	**0.2**	**0.5**	**0.8**	**0.9**	**1.9**	
**Precision (%)**	**6.4**	**4.6**	**5.0**	**3.8**	**2.4**	**2.3**	

**Table 3 pharmaceutics-14-02071-t003:** Intraday precision and accuracy for NAZ in plasma.

	Nominal Concentration (ng/mL)		
	3.0	15.0	75.0
**NAZ**	Measured concentration		
Day 1	2.6	13.3	63.9
	2.8	13.6	66.6
	2.8	14.0	67.3
	3.0	14.4	69.6
	3.1	15.2	70.8
**Mean (5)**	**2.8**	**14.1**	**67.6**
**SD**	**0.2**	**0.7**	**2.7**
**Precision (%)**	**6.4**	**5.3**	**4.0**
**Accuracy (%)**	**94.8**	**94.0**	**90.2**

**Table 4 pharmaceutics-14-02071-t004:** Interday precision and accuracy for NAZ in plasma.

Nominal Concentration (ng/mL)			
	3.0	15.0	75.0
**NAZ**	Measured concentration		
Day 1	2.6	13.3	63.9
	2.8	13.6	66.6
	2.8	14.0	67.3
	3.0	14.4	69.6
	3.1	15.2	70.8
Day 2	2.7	13.2	65.5
	2.7	13.5	67.3
	2.6	12.7	73.0
Day 3	2.7	14.9	65.2
	2.7	13.5	68.5
	2.7	12.9	68.0
Day 4	3.4	16.6	85.7
	3.3	16.7	69.5
	3.3	17.2	90.2
Day 5	3.3	15.4	81.6
	3.3	15.6	83.3
	3.4	15.8	83.9
**Mean (17)**	**3.0**	**14.6**	**72.9**
**SD**	**0.3**	**1.4**	**8.4**
**Precision (%)**	**10.5**	**9.8**	**11.6**
**Accuracy (%)**	**98.6**	**97.5**	**97.2**

**Table 5 pharmaceutics-14-02071-t005:** Intraday precision and accuracy for NAZ in brain tissue.

Nominal Concentration (ng/g)			
	6.0	30.0	60.0
**NAZ**	Measured concentration		
Day 1	6.1	32.7	64.9
	6.4	34.3	67.6
	6.5	33.3	67.5
	6.5	33.4	68.3
	6.2	33.6	67.0
**Mean (5)**	**6.3**	**33.5**	**67.1**
**SD**	**0.2**	**0.6**	**1.3**
**Precision (%)**	**3.3**	**1.7**	**1.9**
**Accuracy (%)**	**105.4**	**111.5**	**111.8**

**Table 6 pharmaceutics-14-02071-t006:** Interday precision and accuracy for NAZ in brain tissue.

Nominal Concentration (ng/g)			
	6.0	30.0	60.0
**NAZ**	Measured concentration		
Day 1	6.1	32.7	64.9
	6.4	34.3	67.6
	6.5	33.3	67.5
	6.5	33.4	68.3
	6.2	33.6	67.0
Day 2	6.7	34.1	64.1
	6.1	33.4	68.5
	6.5	32.7	64.1
Day 3	7.2	28.6	60.3
	6.6	32.7	69.1
	6.4	32.2	68.0
**Mean (17)**	**6.4**	**32.8**	**66.3**
**SD**	**0.2**	**1.5**	**2.7**
**Precision (%)**	**3.5**	**4.7**	**4.0**
**Accuracy (%)**	**106.4**	**109.4**	**110.5**

**Table 7 pharmaceutics-14-02071-t007:** NAZ concentrations in the plasma and brain tissue of healthy mice treated i.p. at a dose of 15 mg/kg (3 times/week for 2 weeks).

Plasma					Brain		
#Mice	Time (h)	NAZ (ng/mL)	Mean (ng/mL)	SD (CV%)	NAZ (ng/g)	MEAN (ng/g)	SD (CV%)
185	Pre-dose	<LOQ	<LOQ	NA	<LOQ	<LOQ	NA
194	Pre-dose	<LOQ			<LOQ		
199	1	347	205.1	133.8 (65.3)	<LOQ	<LOQ	NA
183	1	81.2			<LOQ		
191	1	187			<LOQ		
197	3	5.1	6.7	2.4 (36.3)	<LOQ	<LOQ	NA
198	3	9.4			<LOQ		
192	3	5.5			<LOQ		
182	6	1.2	1.0	0.2 (19.0)	<LOQ	<LOQ	NA
196	6	6.0 #			<LOQ		
184	6	0.9			<LOQ		

# Value outlier not considered due to being pharmacokinetically implausible (i.e., concentration higher at later time points compared to earlier time points.

**Table 8 pharmaceutics-14-02071-t008:** NAZ concentrations in the brain tissue and plasma of grafted and sham-operated mice treated i.p. at a dose of 15 mg/kg (3 times/week for 3 weeks).

Matrix	Sham-Operated Mice			Mice with GSC Tumors	
	#1	#2	#3	#1	#2
**Plasma** (ng/mL)	663	336	1250	2020	2640
Mean ± SD (ng/mL)	750 ± 463			2330 ± 438	
**Brain** (ng/g)	<LOQ	<LOQ	<LOQ	2.6	9.2
Mean ± SD (ng/g)	<LOQ			5.81 ± 4.7	

## Data Availability

Source data are available from the corresponding author upon reasonable request.

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
