# Peer review of "Development and Validation of a HPLC–MS/MS Method to Measure Nifuroxazide and Its Application in Healthy and Glioblastoma-Bearing Mice"

_pharmaceutics, 2022, doi:10.3390/pharmaceutics14102071_

Round 1

Reviewer 1 Report

The authors discuss the quantitation of nifuroxazide in mouse brain and plasma. Advancement of instrumentation has made the quantitation much easier and repurposing of the drug has led to renewed interest. The paper is well written and is of interest to the readers.

One minor comment which can be made is since there are no interfering impurities, the authors can speed up the separation and develop an isocratic method for the quantification which would make method transfer between systems easier.

Also can the authors define how they determined the limit of quantitation mathematically or experimentally)? Since a most of 

Reviewer 2 Report

This work explores the use of a HPLC–MS/MS method in pharmakokinetic monitoring of NAZ. The authors used plasma and brain tissue to test their hypothesis. After carefully reviewing the manuscipt:

1. The abstarct should give more information on why NAZ is investigated

2. Materials and methods should be re-organized. It was very confusing to read first the methods applied and then the moder used

3. The drug was diluted in DMSO. Was there a control administartion just with DMSO in order to be sure about some conclusions of the study?

4. I suggest to compare the method described here with the previous existing with more detail

5. Until now the drug is administered orally. What about its pharmakokinetics to other tissues?

6. At the presentation of the results it would be useful to present the concentration of the drug with the same units at every stage.

7. I dont feel that there is enough data or comments to support the second sentence of the title of the manuscript

Reviewer 3 Report

Manuscript pharmaceutics-1878194 entitled “Development of a HPLC–MS/MS method to measure Nifuroxazide in plasma and brain. Preliminary pharmacokinetic study in glioblastoma bearing mice” developed and validated an assay for quantification Nifuroxazide in mouse plasma and brain. The manuscript is well-written. And there are some points that need to be corrected or clarified before publication:

1.       The title of the manuscript needs to be improved and should not have a punctuation mark of the period.

2.       Regarding all numbers from involved validation studies through the manuscript, please ensure all numbers have the same decimal places.

3.       Line 109: label the model of the centrifuge or use centrifugal force instead of rpm.

4.       Could the author explain why only three days of accuracy and precision results were learned for the brain, but 5 days were spent to learn the plasma’s accuracy and precision results?
